# Role of *Bacillus cereus* in Improving the Growth and Phytoextractability of *Brassica nigra* (L.) K. Koch in Chromium Contaminated Soil

**DOI:** 10.3390/molecules26061569

**Published:** 2021-03-12

**Authors:** Nosheen Akhtar, Noshin Ilyas, Humaira Yasmin, R. Z. Sayyed, Zuhair Hasnain, Elsayed A. Elsayed, Hesham A. El Enshasy

**Affiliations:** 1Department of Botany, PMAS-Arid Agriculture University, Rawalpindi 46300, Pakistan; noshee.nawaz444@gmail.com; 2Department of Biosciences, COMSATS University, Islamabad (CUI), Islamabad 46300, Pakistan; humaira.yasmin@comsats.edu.pk; 3Department of Microbiology, PSGVP Mandal’s Arts, Science, and Commerce College, Shahada, Maharashtra 425409, India; sayyedrz@gmail.com; 4Department of Agronomy, PMAS-Arid Agriculture University, Rawalpindi 46300, Pakistan; zuhair@uaar.edu.pk; 5Bioproducts Research Chair, Zoology Department, Faculty of Science, King Saud University, Riyadh 11451, Saudi Arabia; eaelsayed@ksu.edu.sa; 6Natural & Microbial Products Dept., National Research Centre, Dokki, Cairo 1165, Egypt; 7Institute of Bioproduct Development (IBD), Universiti Teknologi Malaysia (UTM), Skudai, Johor Bahru 81310, Johor, Malaysia; 8City of Scientific Research and Technology Applications, New Burg Al Arab, Alexandria 21934, Egypt

**Keywords:** chromium, *Bacillus cereus*, *Brassica nigra*, heavy metal

## Abstract

Plant growth-promoting rhizobacteria (PGPR) mediate heavy metal tolerance and improve phytoextraction potential in plants. The present research was conducted to find the potential of bacterial strains in improving the growth and phytoextraction abilities of *Brassica nigra* (L.) K. Koch. in chromium contaminated soil. In this study, a total of 15 bacterial strains were isolated from heavy metal polluted soil and were screened for their heavy metal tolerance and plant growth promotion potential. The most efficient strain was identified by 16S rRNA gene sequencing and was identified as *Bacillus cereus*. The isolate also showed the potential to solubilize phosphate and synthesize siderophore, phytohormones (indole acetic acid, cytokinin, and abscisic acid), and osmolyte (proline and sugar) in chromium (Cr^+3^) supplemented medium. The results of the present study showed that chromium stress has negative effects on seed germination and plant growth in *B. nigra* while inoculation of *B. cereus* improved plant growth and reduced chromium toxicity. The increase in seed germination percentage, shoot length, and root length was 28.07%, 35.86%, 19.11% while the fresh and dry biomass of the plant increased by 48.00% and 62.16%, respectively, as compared to the uninoculated/control plants. The photosynthetic pigments were also improved by bacterial inoculation as compared to untreated stress-exposed plants, i.e., increase in chlorophyll a, chlorophyll b, chlorophyll a + b, and carotenoid was d 25.94%, 10.65%, 20.35%, and 44.30%, respectively. Bacterial inoculation also resulted in osmotic adjustment (proline 8.76% and sugar 28.71%) and maintained the membrane stability (51.39%) which was also indicated by reduced malondialdehyde content (59.53% decrease). The antioxidant enzyme activities were also improved to 35.90% (superoxide dismutase), 59.61% (peroxide), and 33.33% (catalase) in inoculated stress-exposed plants as compared to the control plants. *B. cereus* inoculation also improved the uptake, bioaccumulation, and translocation of Cr in the plant. Data showed that *B. cereus* also increased Cr content in the root (2.71-fold) and shoot (4.01-fold), its bioaccumulation (2.71-fold in root and 4.03-fold in the shoot) and translocation (40%) was also high in *B. nigra*. The data revealed that *B. cereus* is a multifarious PGPR that efficiently tolerates heavy metal ions (Cr^+3^) and it can be used to enhance the growth and phytoextraction potential of *B. nigra* in heavy metal contaminated soil.

## 1. Introduction

Heavy metal accumulation is one of the most serious environmental problems in the world that poses negative impacts on soil, plants, animals, and humans [1]. Previously, heavy metal pollution was due to natural sources like volcanic eruption and weathering of metals-bearing rocks or anthropogenic activities like mining and dumping of industrial waste on land. However, in the current scenario industrial waste is the leading cause of soil pollution [2]. Heavy metals (Cr, Hg, Cd, As, Co) cannot be degraded into harmless form and they transfer in the food chain, and their bio-magnification causes toxicity and disturbs the ecosystem [3,4,5]. Chromium (Cr) is one of the toxic heavy metals present in industrial effluents. It exists in different oxidation states; CV (VI) is a human carcinogen whereas Cr (III) is also toxic [6]. Many strategies were introduced to counter the detrimental effects of heavy metals among which phytoextraction is a good approach [7,8]. Phytoextraction is getting attention nowadays as it is an innovative, inexpensive, and non-environmentally disruptive strategy. It is the most realistic approach to remove heavy metals from soil [9]. Some plants are good hyper-accumulators, for example, few members of the family Brassicaceae are more prominent, including *Brassica juncea* (L.) Czern. (Cd) [10], *Brassica napus* L. (Zn, Mn, and Pb) [11], and *Thlaspi caerulescens* (*J*.*Presl & C*.*Presl*) F. K. Mey. [12]. These plants uptake heavy metals actively and have a high bioaccumulation rate in aerial parts. *B. nigra* (black mustard) is among one of the tolerant plants of the Brassicaceae family. It can tolerate a wide variety of environmental conditions. Kaskin et al. [13] reported that when *B. nigra* was grown in Cu-contaminated soil, they actively uptook Cu from the soil, and the bioaccumulation rate of Cu in different plant parts was also high due to bioaccumulation related gene expression in the plant. Bonnet et al. [14] also reported that when *B. nigra* was exposed to combined biotic stresses it triggers the biotic stress-related gene expression in plants against chewing herbivores. It was observed that less research has been carried out on this plant and it is not fully explored for its potential against different heavy metals. On the other hand, many factors restrict the use of other plants for phytoremediation of heavy metals, including (a) reduced growth, (b) phytotoxicity, (c) reduced metal uptake, and (d) less root growth and biomass [15].

To overcome these challenges, inoculation of heavy metal tolerant plant growth-promoting rhizobacteria (PGPR) is getting more attention. They increase stress tolerance in plants and improve their growth significantly. Plant and microbe work synergistically and their interaction results in increased uptake of heavy metals. The basic mechanism includes the production of 1-aminocyclopropane-1-carboxylic acid (ACC) deaminase, siderophore, Indole acetic acid (IAA), phosphate solubilization, and nitrogen fixation [16,17]. It was reported that the inoculation of PGPR enhances the availability of nutrients to plants especially the uptake of nitrogen, phosphorus, and iron was improved significantly. They colonize in the rhizosphere and convert the insoluble bound form of ions into the available form so that they can be absorbed from plant roots and their bioaccumulation was improved [18,19,20]. They have the potential to degrade/solubilize heavy metals by producing metabolizing enzymes and enhance the phytoextraction potential of plants. The optimistic interaction between plant and PGPR enhanced the rate of phytoextraction of heavy metals by plants [21]. Mushtaq et al. [22] reported that inoculation of *B. cereus* enhanced the uptake of Cr, Cd, and Cu in *Catharanthus roseus* (L.) G. Don. The research work of Narayanan et al. [23] showed that *B. cereus* mobilizes heavy metals like Cd, Zn, Mn, and Pb and enhanced their uptake and accumulation in *Jatropha curcas* L. Ke et al. [24] also reported that application of *Bacillus* spp. increased the phytoextraction efficiency of *Lolium perenne* L. in Cu contaminated soil. There is no extensive research work exploring the role of *B. nigra* on phytoextraction of heavy metals. The objectives of the current research were (1) isolation and characterization of efficient strains of PGPR and (2) to evaluate the role of selected strains on growth, metal uptake, translocation, and bioaccumulation in *B. nigra* in Chromium contaminated soil.

## 2. Results

In this research, we screened the phytoremediation potential of *B. nigra* in chromium (Cr) contaminated soil and the role of *B. cereus* inoculation on plant growth and its phytoextraction abilities in the presence of Chromium stress.

### 2.1. Isolation, Heavy Metal Tolerance, and Plant Growth-Promoting Characteristics of Bacterial Strains

The soil sample, isolated from the rhizosphere of *Chenopodium album* L. growing in a heavy metal contaminated site, was used for bacterial isolation. A total of 15 bacterial strains were isolated, based on their morphology. The heavy metal tolerance abilities of these isolates were also tested (Appendix A). It was observed that all strains survive in the presence of 20 mg/L Cr stress while six strains survived in the presence of 30 mg/L Cr stress. When the stress was Cr-40 mg/L, only four strains survived which were further screened for plant growth-promoting characteristics (Appendix A). Out of four Cr tolerant strains, only one strain (B05) was efficient in the production of siderophore, 1-aminocyclopropane-1-carboxylate deaminase, organic acid and has the potential of phosphate solubilization.

### 2.2. Phytohormones and Osmolyte Production by B. cereus in the Presence of Chromium Stress

The bacterial strain was selected based on its stress tolerance and plant growth-promoting characteristics. Data regarding phytohormones (indole acetic acid IAA, cytokinin CK, abscisic acid ABA) and osmolyte (proline, sugar) production was also collected (Table 1). The results showed that bacterial strains facing stress at Cr-20 mg/L synthesized a high amount of phytohormones (IAA 62.66 µg/mL, CK 1.22 µg/mL, ABA 6.48 µg/mL) as compared to the bacterial strains facing the stress of Cr-30 mg/L and Cr-40 mg/L.

Analysis for osmolytes revealed that a high amount of proline (27 µg/mL) was produced at Cr-20 mg/L as compared to other stress levels. The synthesis of sugar was also pronounced in the presence of stress as compared to control. The maximum amount of sugar (43.4 µg/mL) was also observed at Cr-20 mg/L as compared to other stress levels.

### 2.3. Selection and Identification of Bacterial Strain by 16S rRNA Sequencing

The bacterial strain N05 was selected based on its stress tolerance characteristics. It was gram-positive, rod-shaped, and made shiny colonies with irregular margins. The selected bacterial strain (B05) was subjected to molecular characterization (named NA05) based on 16S rRNA sequencing. The strain was identified as *Bacillus cereus* with accession number MW316052 (Appendix A).

### 2.4. Role of B. cereus on Germination of B. nigra

Data collected for germination has been presented in Table 2. It was evident that chromium stress significantly reduced the germination of *B. nigra* (Figure 1). The decrease in germination percentage, seedling length, and vigor index was 39.84%, 55.47%, and 73.70%, respectively, while inoculation of *B. cereus* considerably improved the germination attributes of seeds. The increase in germination percentage by inoculation was 28.07%, 15.78%, and 7.14% with different levels of Cr stress. The maximum increase in seedling length and vigor index was 38.13% and 60.29% by *B. cereus* inoculation in seedlings facing Cr stress.

### 2.5. Role of B. cereus on Plant Growth and Biomass Production

The data of morphological parameters is shown in Table 3. It was noted that Cr stress affects the morphology of the plant and decreases shoot length to 51.57%, 57.05%, and 60.73%, and root length decreased by 48.40%, 19.68%, and 13.04% in the presence of different concentrations of Cr. The decrease in fresh and dry weight was 60.48% and 57.42%, respectively. The inoculation of *B. cereus* improved the shoot length and root length significantly (*p* < 0.05). The increase in shoot length was 35.86%, 19.68%, and 13.04% while root length was increased by 19.11%, 11.73%, and 8.52% as compared to uninoculated stress exposed-plants. The plants exposed to Cr stress have a marked reduction in biomass production which may be due to Cr toxicity. The maximum increase in fresh (48% increase) and dry biomass (62.16%) was observed in plants inoculated with *B. subtilis* as compared to uninoculated stress-exposed plants.

### 2.6. Role of B. cereus on the Content of Photosynthetic Pigments of Plant

Data regarding photosynthetic pigments are shown in Figure 2A,B. It was noted that Cr stress decreased chlorophyll a, chlorophyll b, chlorophyll a + b, and carotenoid contents up to 3.57-fold, 1.98-fold, 2.78-fold, and 1.98-fold, respectively. The inoculation of *B. cereus* considerably increased the content of these photosynthetic pigments. The maximum increase was 25.94%, 10.65%, and 20.35% in chlorophyll a, chlorophyll b, and chlorophyll a + b content, respectively, as compared to uninoculated stress-exposed plants. The increase in carotenoid content was also significant (*p* < 0.05). It was observed that inoculation improved the carotenoid content by 44.30%, 41.75%, and 39.68% in the presence of different levels of Cr stress.

### 2.7. Role of B. cereus on the Water Status of Plant

Chromium stress negatively affected the relative water content (Figure 3) of *B. nigra*, i.e., the decrease in relative water content was 31.68%, 34.31%, and 37.12% in plants facing Cr stress. However, inoculation of *B. cereus* significantly (*p* < 0.05) improved this parameter in the plant. The increase in relative water content was 21.32%, 20.32%, and 17.14% in the presence of different concentrations (20, 30, and 40 mg/kg Cr) of chromium.

### 2.8. Role of B. cereus on Proline and Sugar Content of Plant

Proline is an osmolyte and its concentration was increased in plants facing Cr stress as compared to control (Table 4). It was also reported that proline content in the plant was decreased with the increasing concentration of Cr in the media whereas inoculation of *B. cereus* improved the synthesis and accumulation of proline, i.e., the increase was 8.76%, 8.06%, and 5.50% in the presence of different concentrations of Cr in the media. The synthesis of sugar was decreased in *B. nigra* may be due to Cr toxicity. While inoculation of *B. cereus* increased the synthesis and accumulation of sugar to 28.71%, 17.62%, and 8.75% as compared to uninoculated stress-exposed plants.

### 2.9. Role of B. cereus on Malondialdehyde Content (MDA) and Membrane Stability Index (MSI) of Plant

Malondialdehyde content increased in plants facing Cr stress (Table 5). In this study, Cr stress increased MDA content to 4.15-fold, 5.50-fold, and 7.28-fold as compared to control. It was also observed that the inoculation of *B. cereus* reduced the damaging effects of Cr and decreased malondialdehyde content by 59.53%, 34.28%, and 28.01% as compared to uninoculated stress facing plants. Chromium toxicity decreased membrane stability of *B. nigra* and more damage was reported at Cr-40 mg/kg, i.e., 57.56% decrease. It was observed that inoculation of *B. cereus* increased membrane stability of plant significantly (*p* < 0.05). The increase in membrane stability was 51.39%, 35.21%, and 31.47% as compared to untreated stress-exposed plants.

### 2.10. Role of B. cereus on Antioxidant Enzymes Activities of Plant

To determine the role of *B. cereus* on *B. nigra* facing Cr stress, the data regarding antioxidant enzyme activities was also collected and analyzed (Figure 4). The antioxidant enzyme activities was high at a low concentration of Cr (20 mg/kg) but the increase in stress decreased the level of these enzymes which may be due to increased toxicity of Cr. The inoculation increased superoxide dismutase content by 35.90%, 32.60%, and 29.72% while the increase in peroxide content was 59.61%, 57.82%, and 50.03% in the presence of Cr stress. The maximum increase in catalase content was 33.33% as compared to uninoculated stress-exposed plants.

### 2.11. Role of B. cereus on Uptake, Bioaccumulation, and Translocation of Chromium in Plant

The data of chromium content, its bioaccumulation, and translocation in the plant is shown in Table 6. Chromium uptake and accumulation was high in plants inoculated with *B. cereus* as compared to uninoculated one. The increase in Cr content in root was 2.71-fold, 89.19%, and 36.49%, and in the shoot, the Cr content increased by 4.01-fold, 2.95-fold, and 2.55-fold as compared to uninoculated stress exposed plants. Inoculation of *B. cereus* also improved bioaccumulation and translocation of Cr in plants significantly (*p* < 0.05). The maximum increase in bioaccumulation of Cr in root and shoot and translocation factor was 2.71-fold, 4.03-fold, and 40% as compared to uninoculated stress-exposed plants.

## 3. Discussion

Rhizobacterial strains posses different adaptive mechanisms which make their survival possible in heavy metal contaminated soil. They also have plant growth-promoting traits, so they can be isolated from soil and used for remediation of contaminated soil [25,26]. In this research, heavy metal-polluted soil was screened for bacterial strains and one of the most efficient strains was selected and identified as *B. cereus*. It shows more tolerance toward chromium stress and has plant growth-promoting characteristics. It shows a positive response for siderophore production, 1-aminocyclopropane 1-carboxylic acid (ACC) deaminase synthesis, phosphate solubilization, and organic acid production. In this research, the role of *B. cereus* in improving the growth and phytoextraction abilities of *B. nigra* was investigated.

It was reported in the literature that plant growth-promoting rhizobacteria (PGPR) can enhance the growth of the plants and improve the soil conditions. It was also documented that when plants having heavy metal accumulation abilities were grown for phytoremediation purposes they showed less biomass and had a very slow growth rate [27]. Research studies revealed that rhizobacterial strains having the characteristics of siderophore production, phosphate solubilization, organic acid synthesis, ACC deaminase synthesis, and hydrogen cyanide production abilities have the potential to improve the growth of the plant and can enhance metal uptake in contaminated soil [28]. It was observed that heavy metal stress decreases the growth and yield of economically important crops including wheat [29], rice [30], and maize [31]. Heavy metals directly affect plant metabolism, reduce nutrients and water uptake, disturb cell division, and elongation which results in decreased biomass production. PGPR improves plant growth and increases nutrients uptake and releases phytohormones and other plant growth-promoting substances in the rhizosphere. PGPR release siderophore which improve iron uptake while organic acids increase the phosphorus availability in the rhizosphere [32,33,34]. In this research, the data analysis showed that inoculation of *B. cereus* improved the germination attributes and growth of *B. nigra* in Cr contaminated soil as compared to uninoculated stress exposed plants. Gupta et al. [35] also reported that *Klebsiella sp*. and *Enterobacter sp*. improved the growth parameters of tomato in Cr contaminated soil. Seleiman et al. [36] investigation showed that Cr resistant microbes reduced Cr toxicity and improve the growth of wheat. The production of biomass, availability of heavy metals in the rhizosphere, and their translocation to aerial parts of plants are the important factors affecting phytoremediation. Guo and chi [37] reported that PGPR inoculation increased the biomass of *Lolium multiflorum* Lim. which can be related to IAA and ACC deaminase production. They also increased extractable Cd in the rhizosphere and Cd concentration in the roots and shoots of the plant. Abdollahi et al. [38] research showed that inoculation of PGPR increased the phytoavailability of Cd and Pb and also improved the growth of cabbage varieties. Konkolewska et al. [39] also documented that PGPR increased the rate of heavy metal phytoextraction mainly by increasing the production of dry biomass.

In this study, the analysis of photosynthetic pigments and water status showed that Cr stress imposes a negative effect on these parameters. It may be due to the reason that Cr causes structural changes and damages the photosynthetic machinery of plants, resulting in decreased chlorophyll a, chlorophyll b, total chlorophyll, and carotenoid contents [40]. It was also investigated by Ahmad et al. [41] that Cr stress declined chlorophyll content and gaseous exchange in cauliflower. The main reason was the loss of integrity of the thylakoid membrane. The other heavy metals including Fe, Hg, Zn, Pb, and Cu also effect the normal activities of chlorophyll synthase, rubisco, δ-aminolevulinic acid dehydratase, and protochloro phyllide which are required for chlorophyll synthesis [42]. Our findings are in agreement with Bo et al. [43] who also reported that Cr toxicity reduced the chlorophyll content of maize as compared to control. PGPR release essential substances and increase nutrients availability which plays an important role in the synthesis of photosynthetic pigments which were required for the process of photosynthesis [44]. Zafar-ul-Hye et al. [45] research show that *Stemtrophomonas maltaphilia* and *Agrobacterium fabrum* inoculation increased chlorophyll a chlorophyll b and total chlorophyll content in the bitter gourd in Cd contaminated soil. Plants facing heavy metal stress have reduced water uptake which decreases the photosynthetic capacity of plants and results in a reduced growth rate [46]. It was reported that the application of PGPR decreases the level of ethylene by ACC deaminase production and also increases root growth which enhances water uptake in plants. They also reduce the toxicity of heavy metals and make the plants tolerant in stress conditions [47]. Our findings are in line with Ahmad et al. [48] who also reported that Cd tolerant bacteria improve relative water content in cereals.

Plants try to maintain water potential and osmotic potential in stress conditions. Maintainence of water potential in plants also involves the production of osmolytes like sugar, proline, glycine betaine, sorbitol, and mannitol. Plants that release such osmolytes in stress are more tolerant and it also helps in maintaining cell turgor which is the primary requirement for cell division [49]. PGPR also synthesize osmolytes and phytohormones which maintain cell homeostasis and improve plant growth. In this study, the contents of proline and sugar were increased by *B. cereus* inoculation as compared to untreated stress-exposed plants. The high concentration of heavy metals reduce the synthesis of these osmolytes as Khanna et al. [50] reported that Cd toxicity decreases the content of reducing sugar, glycine betaine, free amino acid, and proline by 65.5%, 59%, 63%, and 54.8%, respectively, in *Solanum lycopersicum* L., while the inoculation of PGPR enhanced osmolytes production and improved stress tolerance in plants. The increase in osmolyte production can be due to the upregulation of genes encoding enzymes for osmolytes synthesis and accumulation. Chromium-induced oxidative stress in plants cause damage to membrane stability and disrupts the DNA, RNA, and proteins. It also causes lipid peroxidation, degrades photosynthetic pigments, and reduces plant growth. It results in the production of reactive oxygen species which decreases membrane stability and damage cell organelles. Chromium toxicity also reduces the rate of antioxidant enzyme activities in plants [51,52]. In this study Cr stress resulted in increased MDA level, decreased MS and antioxidant content which was also lower at high Cr (40 mg/kg) stress in uninoculated plants. It was reported in the literature that inoculation of *B. xiamenesis* PM14 increases POD (6.96%), SOD (30.09%), and CAT (0.89%) activities and decrease MDA content and electrolyte leakage by 27.53% and 2.73%, respectively, in *Sesbania sesban* (L.) Merrill grown in Cr contaminated soil [53]. Our results also show that inoculation of *B. cereus* improves MS, antioxidant activities, and decreases MDA level as compared to uninoculated stress-exposed plants. These results confirmed the role of *B. cereus* in imposing Cr tolerance in plants. These findings have also been reported by Nemat et al. [54] who discussed an increase in ascorbate peroxidase, CAT, SOD POD, and a decrease in MDA and hydrogen peroxide content in *Capsicum annum* L. facing Cr stress. These antioxidant enzymes scavenge the ROS and maintain the cell membrane integrity, biomolecules, and cell organelles structure [55].

In this research, *B. nigra* shows high uptake, bioaccumulation, and translocation of Cr in plants inoculated with *B. cereus* as compared to uninoculated plants. Inoculation of PGPR may alter soil chemistry and enhance the solubility and availability of heavy metal to plant roots thus increased metal uptake and accumulation in roots and shoots [56]. PGPR also release metal degrading enzymes, iron chelators, siderophore, and an organic acid which reduce the toxic effects of heavy metals on plants. They also maintain pH which intern increases nutrient and metal uptake [57]. It was documented that PGPR increase Cd accumulation in *Solanum nigrum* L. [58]. Braud et al. [59] results revealed that heavy metal tolerance and siderophore-producing bacteria increase phytoextraction of Cr and Pb in maize. This study shows that *B. cereus* inoculation increases metal uptake and its accumulation in roots and shoots of *B. nigra*. It was reported that *B. cereus* inoculation reduced the phytotoxicity of heavy metals and enhanced their uptake significantly. The rate of accumulation of heavy metals in roots was high as compared to shoots, which may be due to the accumulation of metals in the vacuoles of root cells and reduces their toxicity. These findings were also reported by Babu et al. [60] who found that inoculation of *Alnus firma* Siebold & Zuccarini with *B. thuringiensis* enhanced the uptake and accumulation of heavy metals in roots. They solubilized the heavy metals by the production of organic acids, metal phosphates, siderophore, and sequester them inside the cells of roots. Ndeddy Aka and Babalola [61] demonstrated that *B. subtilis* increased Ni uptake in the shoot and root by 32% and 55.9% in *B. juncea* (L.) Czern. The inoculation of *B. subtilis* releases organic acids which enhanced the mobility of heavy metals in the soil by making soluble complexes so roots actively uptake heavy metals from the soil. Sharma et al. [62] also reported that *Bacillus* sp. improved the phytoextraction potential of *Phragmites communis*.

## 4. Materials and Methods

### 4.1. Soil Sampling and Analysis

The soil sample was collected from the rhizosphere of *Chenopodium album* L. growing near the main discharge point of industrial effluent located in Rawat industrial area Rawalpindi (3680 m 33°31–15°00 north latitude and 73°15–20°30 east longitude), Pakistan. The soil was collected in a zipper bag and placed in the Icebox and brought to the laboratory. The powdered soil (1 g) was added in (10 mL) HNO_3_ and kept at room temperature for 24 h and then (5 mL) HClO_4_ was added to it. The mixture was heated (500 °C) until the volume was reduced to 3 mL. Distilled water was added to achieve a volume of 25 mL. The sample was filtered and examined by an atomic absorption spectrophotometer (Model GBC 932805). The detail of soil heavy metal analysis [63] is given in Appendix A.

### 4.2. Isolation, Heavy Metal Tolerance, and Plant Growth-Promoting Characteristics of Bacterial Strains

Isolation of bacterial strains was carried out by the serial dilution method. Approximately 1 g soil was added in 9 mL distilled water and serial dilution was prepared (up to 10^−8^). The strains were plated on nutrient agar plates supplemented with K_2_Cr_2_O_7_ (10 mg/L). The tolerant bacterial colonies were observed after 24 h and 48 h and separated by the streak plate method. The strains were grown in nutrient broth (LB) having 20, 30, and 40 mg/L K_2_Cr_2_O_7_ in an incubator shaker, and their growth was observed by measuring optical density after 24 h and 72 h [64]. Analysis of plant growth-promoting characteristics of bacterial strains was also carried out. Siderophore production was measured by spot inoculation of bacterial strains (10^8^ CFU/mL) on chrome azurol S agar plates. The plates were incubated at 28 °C for 72 h. The development of orange-yellow halo around the growth zone of bacterial strains indicated the production of siderophore [65]. The 1-aminocyclopropane 1-carboxylate deaminase activity was measured by the production of α-ketobutyrate from the cleavage of ACC [66]. The bacterial strains were grown in tryptic soy broth media at 28 °C. When the cells reached the stationary phase, they were collected by centrifugation and washed with 0.1 M Tris-HCl. The cells were suspended in 2 mL of DF (Dworkin and Foster) minimal medium having ACC with and without PEG 6000 and placed on an incubator shaker at 28 °C for 72 h. The rate of production of α-ketobutyrate was analyzed by comparing the absorbance (taken at 540 nm by spectrophotometer) with the standard curve. To measure phosphate solubilization, pikovskaya’s agar plates were prepared having 2% of tricalcium phosphate. The overnight grown bacterial culture (5 µL) was spotted on pikovskaya’s agar plates, incubated at 28 °C for 48 h, and observed for the appearance of solubilization zone which indicated the phosphate solubilization by bacterial strains [67]. The organic acid production was measured by high-performance liquid chromatography (Agilent 1100, Waldbronn, Germany) having a C18 column (39 × 300 mm) and a UV detector. The content was measured by a comparison of chromatogram retention time and peak areas with the standards [68].

### 4.3. Analysis of Phytohormones and Osmolyte Production

Analysis of phytohormones (indole acetic acid, cytokinin, abscisic acid) production of bacterial strain under stress (Cr-20, 30, 40 mg/L) and non-stress (Cr-0 mg/L) conditions was also carried out. Bacterial cultures (200 mL) were centrifuged for 2000× *g* for 20 min, pH (2.8 was adjusted with 1N HCl. The extraction was carried out with ethyl acetate and the resultant solution was evaporated (35 °C) and mixed with methanol (500 µL). The samples were run on high-performance liquid chromatography (HPLC) and blank (without inoculum) Luria Broth media was used for normalizing the data. The chemical hormones (Sigma Chemical Co., St. Louis, MO, USA) were used for the standardization of HPLC (Agilent 1100, Waldbronn, Germany) [69].

Proline and sugar analysis was carried out by culturing the bacterial strain in Luria Broth (LB) media supplemented with Cr-0 mg/L, Cr-20 mg/L, Cr-30 mg/L, and Cr-40 mg/L. The samples were centrifuged at 1000× *g* for 10 min and the supernatant was collected. Approximately 1 mL glacial acetic acid and 1 mL acid ninhydrin were added to the supernatant followed by boiling for 1 h and the reaction was stopped immediately by placing the tubes in the ice bath. The extraction was done with toluene (2 mL) and the organic phase was used to measure the absorbance at 515 nm by spectrophotometer (Model 5520XR, Wescor Inc., Logan, UT, USA) [70]. Soluble sugar was measured by mixing 1 mL of supernatant with 4 mL anthrone reagent and placed in a boiling water bath for 8 min. Optical density was measured after rapid cooling and the standard curve was used for calculation [71].

### 4.4. Identification of Bacterial Strain by 16S rRNA Sequencing

The bacterial strains growing in liquid media (LB) were centrifuged (model 2-16, Sigma, Osterode am Harz, Germany) (1000× *g*) to obtain a pellet and frozen in liquid nitrogen. The pellet was placed in a mortar having lysis buffer, 1.4 M NaCl, 100 mM Tris-HCl, 1% polyvinylpyrrolidone, 0.2% LiCl, and 20 mM disodium salt of ethylenediamine tetra-acetic acid. Approximately 800 µL solutions were added to the sample and after mixing vigorously incubated at 65 °C for 2 h. Samples were centrifuged at 6160× *g* and the supernatant was collected in a separate tube. Equal volume of chloroform-isoamyl alcohol was added and mixed in the supernatant. The sample was centrifuged at 8870× *g*. The upper phase was collected and DNA was precipitated by adding an equivalent volume of isopropanol and stored at −20 °C for 3 min. Centrifugation was done at 12,073× *g* to collect the DNA. The pellet was washed with (500 µL) 96% and 70% ethanol. The pellet was dried at room temperature and dissolved in 50 µL of Tris-EDTA buffer and stored at −20 °C [72]. The DNA of bacterial strain was amplified by a thermocycler having forward primer (fd1) AGAGTTTGATCCTGGCTCAG, and reverse primer (rd1) (AAGGAGGTGATCCAGCC). DNA was sequenced by 16S rRNA gene sequencing in a sequencer. The data was submitted to the National Centre for Biotechnology Information (NCBI) where the sequence was compared with the already sequenced strains to find homology [73]. The strain was identified as *Bacillus cereus* with Accession number MW316052.

### 4.5. Germination Experiment

Germination experiment was carried out in the Plant Physiology laboratory at PMAS Arid Agriculture University Rawalpindi, Pakistan. Seeds of *B. nigra* were collected from the National Agriculture Research Centre (NARC) Islamabad, Pakistan. Seeds were surface sterilized with 0.5% sodium hypochlorite (5 min) and then washed with distilled water. The bacterial strain *B. cereus* used in this experiment was identified by 16S rRNA sequencing having chromium tolerance and plant growth-promoting characteristics. It was grown in LB media (incubator shaker) for 48 h at 28 °C. They were centrifuged at 1000× *g* for 15 min and mixed with distilled water; the final concentration was adjusted at 10^8^ CFU/mL by adjusting the optical density at 1. Seeds were primed with *B. cereus* for 6 h at 28 °C. Control seeds were treated with distilled water [74]. Seeds (twenty) were placed on petri plates having filter paper. The solutions of K_2_Cr_2_O_7_ were prepared with concentrations of 20 mg/L, 30 mg/L, and 40 mg/L. Chromium stress was given by applying (20, 30, and 40 mg/L) 8 mL solution of K_2_Cr_2_O_7_ and control received only distilled water. The experimental design was a completely randomized design with three replicates. Germination parameters like germination percentage, seedling length, and vigor index were taken after 15 days of germination [75]. The formula used was as follows:Germination percentage = Number of seeds germinated/Total number of seeds × 100,
Vigor index = Germination index × Seedling biomass.

### 4.6. Pot Experiment

A pot experiment was carried out in the glasshouse of the Department of Botany at PMAS Arid Agriculture University Rawalpindi, Pakistan. The soil was collected from a local nursery and its pH, electrical conductivity, and texture was measured by following the protocol of Gee and Bauder [76]. Soil pH and electrical conductivity were measured with soil pH meter (Model 3305, Jenway, Mainland, UK) and EC meter (Model WTW 82,362 InoLan, Weilheim, Germany), respectively. Approximately 1 g soil sample was added in 100 mL distilled water and stirred for 5 min. The sample was kept overnight at room temperature and stirred again, allowed to sit for 15 min. The sample was strained and readings were taken. The organic carbon in the soil sample was extracted by 0.5 M K_2_SO_4_. Nutrients like potassium, nitrogen, iron, and phosphorus content were also analyzed by following the standard protocols. To measure chromium content, 2 g soil sample was mixed with (10 mL) 1:1 HNO_3_/H_2_O and heated (10 min) on a digestion block. In 5 mL of solution conc. HNO_3_ was added and heated to complete the oxidation of the sample. The digestion liquid was heated (95 °C) until it was reduced to 5 mL afterwards, and 2 mL of 30% H_2_O_2_ and 2 mL of water were added and again heated to reduce to 5 mL. The sample was cooled to room temperature and 10 mL of HCl was added followed by heating (15 min) at 95 °C. The digests were filtered and diluted with 100 mL of distilled water and analysis was done by atomic absorption spectrophotometer (Model GBC 932805). The results were compared with standards [77]. The soil texture was loam having 7.2 pH, 0.65 dsm^−1^ electrical conductivity, 1.43% organic matter content, and 46.72% saturation (Table 7). The content of available chromium in the soil was 2.68 mg/kg. Method of seeds collection and sterilization were the same as mentioned in Section 2.5 (germination section). The experiment was carried out at the start of October with the temperature range of 15 °C to 23 °C and 1.6 mm precipitation. Pots were filled with 6 kg soil and 120 mL of bacterial culture having 10^8^ CFU/mL was inoculated in the soil and 20 days past incubation ten seeds were sown in each pot. In this experiment, each treatment has three replicates. After one week of germination thinning was done and four plants per pot were maintained. The solutions of K_2_Cr_2_O_7_, i.e., 20 mg/L, 30 mg/L, and 40 mg/L, were prepared and applied at the rate of 500 mL solution per pot. After 10 days of imposing stress (30 days old plant), samples were collected for analysis.

### 4.7. Growth Parameters

Plants were uprooted and their root length and shoot length was measured by using measuring tape. To measure biomass, plants were cleaned to remove soil and weight was taken on measuring balance. After that, they were placed in an oven at 60 °C for 24 h and dry weight was measured [78].

### 4.8. Chlorophyll and Carotenoid Content

Photosynthetic pigments were measured by grinding 0.5 g fresh leaf in 10 mL of 80% acetone. The homogenate was filtered and the absorbance of the filtrate was taken at 645 nm, 663 nm, and 470 nm by spectrophotometer (Model 5520XR, Wescor Inc, Logan, UT, USA) [79]. The following formulas were used for calculation [80,81]:Chla (mg/g) = [12.7A663−2.69A645] (*v*/*w*),
Chlb (mg/g) = [22.9A645−4.68A663] (*v*/*w*),
Carotenoids content (mg/g) = (1000 A470 − 1.8 Chla − 85.02 Chlb)/198.

### 4.9. Estimation of Relative Water Content

Relative water content was measured by following the protocol given by Whetherley [82]. Fresh weight (FW) of leaves were measured by using weighing balance and after that, they were dipped in water for 24 h and their hydrated weight (HW) was measured. They were oven-dried for 24 h at 80 °C and dry weight (DW) was taken. Relative water content was calculated by using the formula
Relative water content = (FW − DW/HW − DW)/100.

### 4.10. Estimation of Proline and Sugar Content

Proline content was measured by following the protocol of Bates [83]. Fresh leaf sample (0.5 g) was homogenized in 10 mL sulfosalicylic acid. The homogenate was filtered and 2 mL filtrate was added in 2 mL ninhydrin reagent and 2 mL glacial acetic acid. The mixture was heated in the water bath for 1 h (90 °C) and cooled by placing it in the ice bath. The sample was transferred in a separating funnel and 4 mL toluene was added to it. The upper layer was isolated and the absorbance was taken at 520 nm by using a spectrophotometer. A standard curve was prepared by using proline.

Leaf (0.1 g) was ground and mixed with 80% methanol and heated in a water bath at 70 °C for 30 min. An equal volume of extract (2 mL) and (4%) phenol was mixed with sulphuric acid (2 mL) and placed in the dark for 30 min. The absorbance was taken at 490 nm by spectrophotometer (Model 5520XR, Wescor Inc., Logan, UT, USA) and a standard curve was also prepared by using glucose [84].

### 4.11. Malondialdehyde Content (MDA) and Membrane Stability Index (MSI)

Malondialdehyde content was measured by following the protocol of Cakmak and Horst [85]. The leaf sample (0.5 g) was homogenized with (2 mL) 0.1% trichloro acetic acid (TCA). The supernatant was collected after centrifugation at 1000× *g* for 10 min and mixed (1 mL) with 4 mL solution having 20% TCA and 0.5% thiobarbituric acid. The mixture was kept in a boiling water bath (30 min) and cooled by placing it in the ice bath. After centrifugation at 1000 *g* for 15 min, the absorbance of the supernatant was measured at 532 nm and 600 nm.

To measure membrane stability, fresh leaves were washed with distilled water and were added to a test tube having distilled water and were kept in a water bath for 30 min at 40 °C. After that electrical conductivity (C1) of leaves were taken. The same samples were again placed in a water bath at 100 °C for 10 min and electrical conductivity (C2) was recorded [86]. Membrane stability was calculated by using the formula
Membrane Stability Index (%) = [1 − C1/C2] × 100.

### 4.12. Antioxidant Enzymes Activities

Enzyme extract was prepared by grinding the sample (1 g) in 10 mL extraction buffer (50 mM potassium (K)-phosphate buffer with 1% polyvinyl polypyrrolidone, pH 7). The homogenate was centrifuged at 15,000 g for 15 min and the supernatant was collected and used for antioxidant enzymes assay [87]. Superoxide dismutase (SOD) activity was measured by following the method of Beauchamp and Fridovich [88]. The reaction mixture (0.75 mM nitroblue tetrazolium, 130 mM methionine, 0.05 M phosphate buffer pH 7, 0.02 mM riboflavin) was mixed with 300 µL enzyme extract. The blank was composed of the same amount of reaction mixture containing extra buffer instead of enzyme extract. Blank and reaction mixture was placed under a fluorescent lamp for 7 min and the absorbance was measured at 560 nm. The superoxide dismutase activity was measured by observing the inhibition in the photoreduction of nitroblue tetrazolium. Peroxidase activity was measured by following the method of Lagrimin [89]. The increase in the absorbance was due to tetraguaiacol formation. Total 10 readings were taken with a 20 s interval at 470 nm. Catalase activity was measured by following the protocol of Aebi [90]. The decrease in absorbance was noticed for 3 min at 240 nm. The reaction mixture (1 mL) was prepared by mixing (250 µL) enzyme extract, 10 × 50 mM K-phosphate buffer, DH_2_O (450 µL), 10 × 60 nM H_2_O_2_ (100 µL). The blank was also prepared to have the same reaction mixture with an additional 250 µL K-phosphate buffer instead of enzyme extract. To stop the reaction, H_2_O_2_ was added at the end and readings were taken for 3 min.

### 4.13. Determination of Chromium Its Bioaccumulation and Translocation

Dried ground plant sample (0.6 g) was mixed with 5 mL HNO_3_ and was placed overnight and 0.5 mL H_2_O_2_ was added and ashed in a muffle furnace for (155 °C) 4.5 h. Later, it was dissolved in distilled water to make the final volume 30 mL and the absorption of the digested solution was measured by atomic absorption spectrophotometer (Model GBC 932805) [91]. Bioaccumulation of chromium in plant and its translocation was calculated by using formulas as given below [92].
Transcription factor = Metal in shoot DW/Metal in root DW,
Bioconcentration factor = Metal in plant part (root or shoot) (DW)/Metal in soil DW.

### 4.14. Statistical Analysis

Statistical analysis was carried out by using the statistix 8.1 software. Analysis of variance was carried out and the design was a completely randomized design (CRD). Each treatment had three replicates. Differences between the mean of treatments were analyzed by Least Significant Difference Test (LSD).

## 5. Conclusions

The present study explores the role of *B. cereus* on plants growing in contaminated soil. It was observed that *B. cereus* improved germination and growth attributes significantly and reduced the toxic effects of chromium stress in *B. nigra*. The content of photosynthetic pigments and the water status of the plant was also improved. Membrane stability was increased as the antioxidant enzymes work more efficiently and their content was also increased. The increase in Cr uptake by plant parts was considerably high due to *B. cereus* inoculation as compared to uninoculated stress-exposed plants. The bioaccumulation and translocation of metal were also significant. Based on our findings, it can be concluded that *B. cereus* is a valuable tool and can be used in the future for the development of contaminants (heavy metals) specific bio-inoculants which also improves the phytoextraction potential of plants.

## Figures and Tables

**Figure 1 molecules-26-01569-f001:**
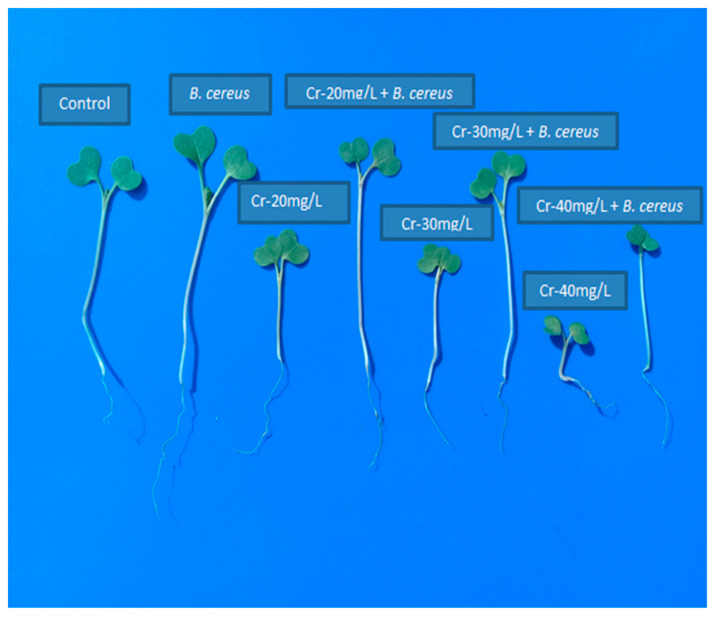
Effect of chromium stress and inoculation of *B. cereus* on germination of *B. nigra.* Where T0 = Control, T1 = *B. cereus*, T2 = Cr-20 mg/kg, T3 = Cr-20 mg/kg + *B. cereus*, T4 = Cr-30 mg/kg, T5 = Cr-30 mg/kg + *B. cereus*, T6 = Cr-40 mg/kg, T7 = Cr-40 mg/kg + *B. cereus*.

**Figure 2 molecules-26-01569-f002:**
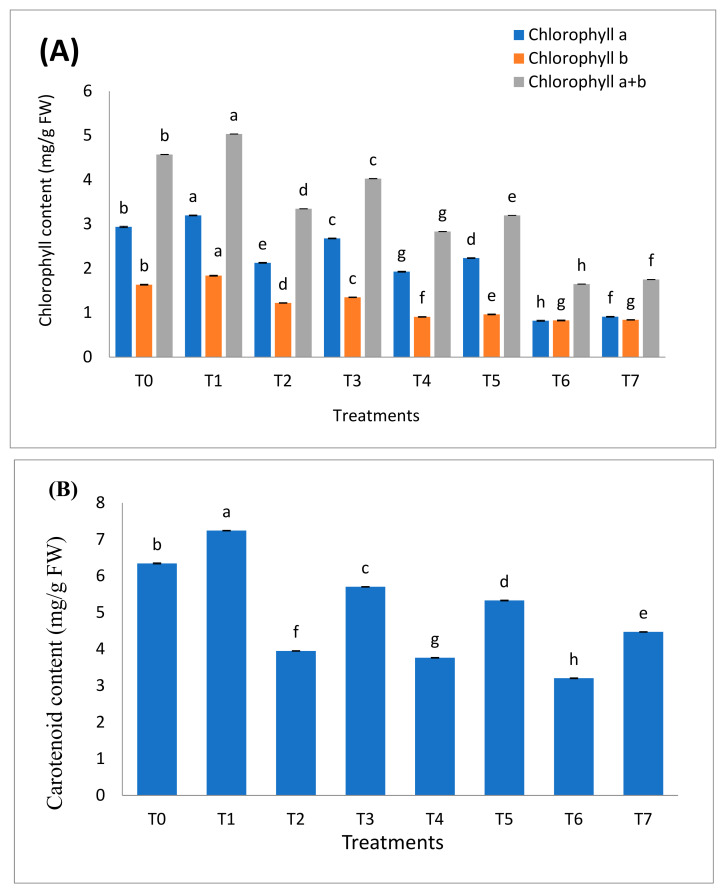
Effect of *B. cereus* inoculation on (**A**) chlorophyll content and (**B**) carotenoid content of *B. nigra*. facing chromium stress. Where T0 = Control, T1 = *B. cereus*, T2 = Cr-20 mg/kg, T3 = Cr-20 mg/kg + *B. cereus*, T4 = Cr-30 mg/kg, T5 = Cr-30 mg/kg + *B. cereus*, T6 = Cr-40 mg/kg, T7 = Cr-40 mg/kg + *B. cereus*. Mean and standard deviations are shown in the figure. Different letters indicate significant differences between means of treatments at *p* < 0.05 according to LSD test.

**Figure 3 molecules-26-01569-f003:**
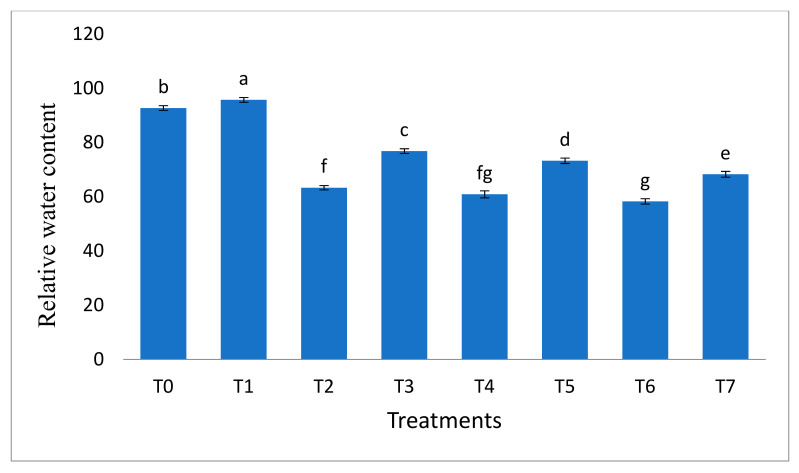
Effect of bacillus inoculation on the relative water content of *B. nigra*. facing chromium stress. Where T0 = Control, T1 = *B. cereus*, T2 = Cr-20 mg/kg, T3 = Cr-20 mg/kg + *B. cereus*, T4 = Cr-30 mg/kg, T5 = Cr-30 mg/kg + *B. cereus*, T6 = Cr-40 mg/kg, T7 = Cr-40 mg/kg + *B. cereus*. Mean and standard deviations are shown in the figure. Different letters indicate significant differences between means of treatments at *p* < 0.05 according to the LSD test.

**Figure 4 molecules-26-01569-f004:**
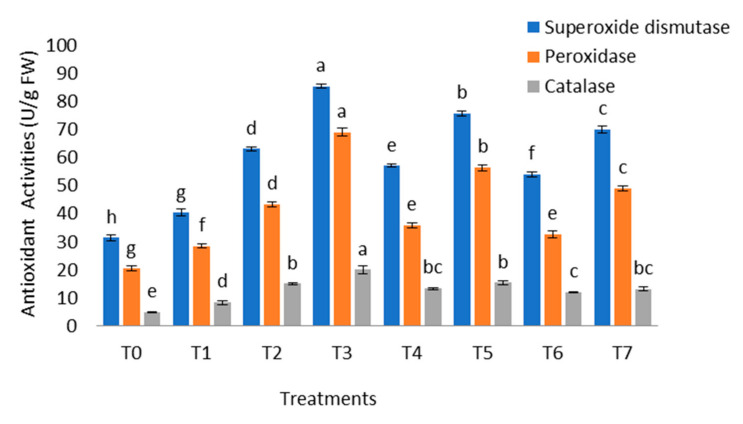
Effect of bacillus inoculation on antioxidant activities of *B. nigra* L. facing chromium stress. Where T0 = Control, T1 = *B. cereus*, T2 = Cr-20 mg/kg, T3 = Cr-20 mg/kg + *B. cereus*, T4 = Cr-30 mg/kg, T5 = Cr-30 mg/kg + *B. cereus*, T6 = Cr-40 mg/kg, T7 = Cr-40 mg/kg + *B. cereus*. Mean and standard deviations are shown in the figure. Different letters indicate significant differences between means of treatments at *p* < 0.05 according to the LSD test.

**Table 1 molecules-26-01569-t001:** Effect of chromium stress on phytohormones and osmolytes production by *Bacillus cereus* grown in the culture medium.

Treatments	IAA (µg/mL)	CK (µg/mL)	ABA (µg/mL)	Proline (µg/mL)	Sugar (µg/mL)
Cr-0 mg/L	81.33 ± 1.76 ^a^	1.55 ± 0.01 ^a^	2.57 ± 0.10 ^c^	10.16 ± 0.35 ^d^	12.76 ± 0.23 ^d^
Cr-20 mg/L	62.66 ± 0.88 ^b^	1.22 ± 0.03 ^b^	6.48 ± 0.20 ^a^	27.00 ± 1.15 ^a^	43.40 ± 1.22 ^a^
Cr-30 mg/L	55.00 ± 1.73 ^c^	0.81 ± 0.00 ^c^	5.84 ± 0.56 ^a^	22.66 ± 1.20 ^b^	34.00 ± 1.52 ^b^
Cr-40 mg/L	48.66 ± 2.02 ^d^	0.62 ± 0.00 ^d^	4.38 ± 0.07 ^b^	17.66 ± 1.45 ^c^	19.66 ± 1.45 ^c^

Values are mean of replicates, standard deviations are also calculated and different letters indicate significant differences between means of treatments at *p* < 0.05 according to LSD test.

**Table 2 molecules-26-01569-t002:** Effect of chromium stress and *B. cereus* inoculation on germination attributes of *Brassica nigra*.

Treatments	Germination Percentage (%)	Seedling Length (cm)	VI
T0	85.33 ± 1.45 ^b^	8.13 ± 0.42 ^b^	692.83 ± 25.32 ^b^
T1	92.33 ± 0.88 ^a^	10.43 ± 0.72 ^a^	964.20 ± 73.68 ^a^
T2	63.00 ± 1.15 ^d^	4.72 ± 0.00 ^de^	297.76 ± 4.91 ^de^
T3	73.00 ± 1.52 ^c^	6.52 ± 0.49 ^c^	477.29 ± 45.64 ^c^
T4	57.00 ± 1.00 ^e^	4.22 ± 0.00 ^ef^	240.74 ± 4.60 ^ef^
T5	66.00 ± 0.57 ^d^	5.74 ± 0.01 ^cd^	379.26 ± 2.48 ^d^
T6	51.33 ± 0.66 ^f^	3.62 ± 0.01 ^f^	185.84 ± 2.98 ^f^
T7	55.00 ± 0.57 ^e^	4.24 ± 0.01 ^ef^	233.21 ± 3.08 ^ef^

Where T0 = Control, T1 = *B. cereus*, T2 = Cr-20 mg/kg, T3 = Cr-20 mg/kg + *B. cereus*, T4 = Cr-30 mg/kg, T5 = Cr-30 mg/kg + *B. cereus*, T6 = Cr-40 mg/kg, T7 = Cr-40 mg/kg + *B. cereus*. Values are mean of replicates, standard deviations is also calculated, and different letters indicate significant differences between means of treatments at *p* < 0.05 according to LSD test.

**Table 3 molecules-26-01569-t003:** Effect of chromium stress and *B. cereus* inoculation on plant growth and biomass of *B. nigra* L.

Treatments	Shoot Length (cm)	Root Length (cm)	Fresh Weight (mg)	Dry Weight (mg)
T0	19.00 ± 1.52 ^b^	15.70 ± 0.95 ^b^	1016.00 ± 21.07 ^b^	35.00 ± 1.15 ^b^
T1	21.70 ± 1.37 ^a^	18.40 ± 0.55 ^a^	1287.70 ± 16.75 ^a^	47.66 ± 0.88 ^a^
T2	9.20 ± 0.73 ^d^	7.50 ± 0.20 ^d^	533.33 ± 16.49 ^e^	19.03 ± 0.38 ^ef^
T3	12.50 ± 0.81 ^c^	8.93 ± 0.088 ^c^	789.33 ± 14.49 ^c^	30.86 ± 0.87 ^c^
T4	8.16 ± 0.72 ^d^	7.13 ± 0.23 ^d^	472.67 ± 6.17 ^f^	17.06 ± 0.24 ^fg^
T5	9.76 ± 0.35 ^d^	7.96 ± 0.12 ^cd^	681.67 ± 14.14 ^d^	27.40 ± 0.37 ^d^
T6	7.46 ± 0.55 ^d^	7.00 ± 0.05 ^d^	402.33 ± 15.49 ^g^	14.90 ± 0.40 ^g^
T7	8.23 ± 0.34 ^d^	7.59 ± 0.04 ^d^	571.33 ± 12.60 ^e^	20.33 ± 1.24 ^e^

Where T0 = Control, T1 = *B. cereus*, T2 = Cr-20 mg/kg, T3 = Cr-20 mg/kg + *B. cereus*, T4 = Cr-30 mg/kg, T5 = Cr-30 mg/kg + *B. cereus*, T6 = Cr-40 mg/kg, T7 = Cr-40 mg/kg + *B. cereus*. Values are mean of replicates, standard deviations is also calculated and different letters indicate significant differences between means of treatments at *p* < 0.05 according to LSD test.

**Table 4 molecules-26-01569-t004:** Effect of chromium stress and *B. cereus* inoculation on osmolyte content of *B. nigra.*

Treatments	Proline µg/mg	Sugar µg/mg
T0	1.56 ± 0.01 ^f^	151.33 ± 3.17 ^b^
T1	3.20 ± 0.01 ^e^	185.00 ± 1.73 ^a^
T2	6.50 ± 0.02 ^b^	69.66 ± 1.45 ^d^
T3	7.07 ± 0.03 ^a^	89.66 ± 0.88 ^c^
T4	5.33 ± 9.02 ^d^	58.66 ± 1.45 ^e^
T5	5.76 ± 0.01 ^c^	69.00 ± 1.52 ^d^
T6	5.09 ± 0.33 ^d^	53.33 ± 1.45 ^f^
T7	5.37 ± 0.01 ^d^	58.00 ± 1.15 ^ef^

Where T0 = Control, T1 = *B.*, T2 = Cr-20 mg/kg, T3 = Cr-20 mg/kg + *B. cereus*, T4 = Cr-30 mg/kg, T5 = Cr-30 mg/kg + *B. cereus*, T6 = Cr-40 mg/kg, T7 = Cr-40 mg/kg + *B. cereus*. Values are mean of replicates, standard deviations are also calculated, and different letters indicate significant differences between means of treatments at *p* < 0.05 according to the LSD test.

**Table 5 molecules-26-01569-t005:** Effect of chromium stress and *B. cereus* inoculation on malondialdehyde content and membrane stability index of *B. nigra*.

Treatments	MDA (µmol/g^2^FW)	MSI (%)
T0	4.33 ± 0.01 ^f^	84.60 ± 0.98 ^b^
T1	2.64 ± 0.00 ^g^	91.86 ± 1.21 ^a^
T2	18.00 ± 0.32 ^c^	43.00 ± 0.87 ^f^
T3	7.32 ± 0.01 ^e^	65.10 ± 0.88 ^c^
T4	23.83 ± 0.64 ^b^	38.90 ± 1.30 ^g^
T5	15.66 ± 0.59 ^d^	52.60 ± 0.85 ^d^
T6	31.56 ± 0.67 ^a^	35.90 ± 1.26 ^g^
T7	22.70 ± 0.51 ^b^	47.20 ± 0.72 ^e^

Where T0 = Control, T1 = *B. cereus*, T2 = Cr-20 mg/kg, T3 = Cr-20 mg/kg + *B. cereus*, T4 = Cr-30 mg/kg, T5 = Cr-30 mg/kg + *B. cereus*, T6 = Cr-40 mg/kg, T7 = Cr-40 mg/kg + *B. cereus*. Values are mean of replicates, standard deviations are also calculated, and different letters indicate significant differences between means of treatments at *p* < 0.05 according to the LSD test.

**Table 6 molecules-26-01569-t006:** Effect of chromium stress and *B. cereus* inoculation on chromium uptake, its bioconcentration, and translocation in *B. nigra*.

Treatments	Root Cr Content (mg/g DW)	Shoot Cr Content (mg/g DW)	Bioconcentration Factor (Root)	Bioconcentration Factor (Shoot)	Translocation Factor
T0	1.54 ± 0.00 ^g^	0.41 ± 0.01 ^f^	0.57 ± 0.00 ^f^	0.13 ± 0.02 ^g^	0.26 ± 0.00 ^a^
T1	2.36 ± 0.01 ^g^	0.67 ± 0.00 ^f^	0.88 ± 0.00 ^f^	0.24 ± 0.00 ^g^	0.28 ± 0.00 ^a^
T2	314.67 ± 4.91 ^f^	31.70 ± 1.00 ^e^	15.73 ± 0.24 ^e^	1.58 ± 0.05 ^f^	0.10 ± 0.00 ^e^
T3	855.00 ± 10.44 ^d^	127.33 ± 12.34 ^c^	42.75 ± 0.52 ^a^	6.36 ± 0.61 ^c^	0.14 ± 0.01 ^c^
T4	607.67 ± 12.66 ^e^	75.30 ± 1.73 ^d^	19.75 ± 0.91 ^d^	2.50 ± 0.05 ^e^	0.12 ± 0.00 ^d^
T5	1149.70 ± 8.08 ^b^	222.67 ± 7.68 ^b^	35.11 ± 3.14 ^b^	7.42 ± 0.25 ^b^	0.19 ± 0.00 ^b^
T6	960.67 ± 10.10 ^c^	139.67 ± 7.31 ^c^	24.01 ± 0.25 ^c^	3.49 ± 0.18 ^d^	0.14 ± 0.00 ^cd^
T7	1311.30 ± 15.98 ^a^	356.17 ± 6.20 ^a^	32.78 ± 0.39 ^b^	8.90 ± 0.15 ^a^	0.27 ± 0.00 ^a^

Where T0 = Control, T1 = *B. cereus*, T2 = Cr-20 mg/kg, T3 = Cr-20 mg/kg + *B. cereus*, T4 = Cr-30 mg/kg, T5 = Cr-30 mg/kg + *B. cereus*, T6 = Cr-40 mg/kg, T7 = Cr-40 mg/kg + *B. cereus*. Values are mean of replicates, standard deviations is also calculated, and different letters indicate significant differences between means of treatments at *p* < 0.05 according to LSD test.

**Table 7 molecules-26-01569-t007:** Physicochemical analysis of soil used in the experiment.

Texture	Loam
pH	7.20 ± 0.24
Electrical conductivity (dsm^−1^)	0.65 ± 0.00
Organic matter (%)	1.43 ± 0.03
Saturation (%)	46.72 ± 1.68
Potassium (mg/kg)	132.29 ± 7.42
Nitrogen (mg/kg)	61.03 ± 2.36
Iron (mg/kg)	4.76 ± 0.06
Phosphorus (mg/kg)	5.34 ± 0.02
Bioavailable Cr (mg/kg)	2.68 ± 0.04

## Data Availability

All the data are available in the manuscript and its Appendix A.

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
