# Peer review of "Role of Bacillus cereus in Improving the Growth and Phytoextractability of Brassica nigra (L.) K. Koch in Chromium Contaminated Soil"

_molecules, 2021, doi:10.3390/molecules26061569_

Round 1

Reviewer 1 Report

The work "Role of Bacillus subtilis in improving the growth and phytoextractability of Brassica nigra L. in chromium contaminated soil" concerns the important role of PGPR bacteria in increasing plant tolerance to metal stress. Unfortunately, the work contains numerous stylistic, editorial and substantive errors and in that form can not be accepted.

English must be corrected by Native Speaker.

Abstract:

  • correct unnecessary double spaces; Cr ions are cations, not anions,
  • line 38 - catalase from lowercase
  • line 92 - chrome also in lower case
  • lines 68, 146 - the names of plants and bacteria in Latin are written in italics
  • lines 134,269 - wrong spelling, it should be 16S rRNA
  • line 129 - expand NBT and other abbreviations in the text
  • line 159 - what are the micro and macro elements?
  • line 163 - incorrectly written kg unit; 20 days of incubation ?? or days past inoculation
  • lines 377, 239 - all lowercase
  • line 196 - the absorbance
  • line 212 - no spaces
  • line 215 - dot not needed
  • line 221 - no comma, etc.
  • unclear - from line 223 - description unclear, 253-258 the same

Methods - the description of the methods is imprecise, no description of the methods used, e.g. for the determination of PGP activity,

The entire methodology must be corrected.

- Why in the pot experiment the chromium was used only after 30 days and not immediately when sowing,as in the plates experiment

- lines 206,222 - what was the plant sample? it is necessary to specify whether they were aerial parts or whole plants

- no description of the method for determining the concentrations of metals, are they total or mobile values ?? and what was the pH of the soil. It is known that metal ions become mobile at acidic pH 

Author Response

RESPONSE TO FIRST REVIEWR’S COMMENTS

AUTHOR’S RESPONSE: The authors are thankful for the valuable suggestion and comments of the editorial team which helped in the improvement of the manuscript. The whole manuscript has been revised keenly in the light of suggestions proposed by reviewers. Your valuable suggestions are well taken to improve the MS.  The point wise response to each of your comment/suggestions is given below.

Abstract:

  • correct unnecessary double spaces; Cr ions are cations, not anions

AUTHOR’S RESPONSE: whole manuscript has been checked thoroughly and extra double spaces has been removed. L 27 and 43 (page 1), corrected as Cr+3.

  • line 38 - catalase from lowercase

AUTHOR’S RESPONSE: L 38 (page 1), catalase has been written in lower case.

  • line 92 - chrome has been written in in lower case

AUTHOR’S RESPONSE: L 100 (page 3) chrome has been written in lower case

  • lines 68, 146 - the names of plants and bacteria in Latin are written in italics

AUTHOR’S RESPONSE: The plant and bacterial names have been written in italics throughout the text.

  • lines 134,269 –wrong spelling, it should be 16S rRNA

AUTHOR’S RESPONSE: L 137-315, (page 4, 5 and 9) Correction has been done as 16S rRNA in the manuscript.

  • line 129 - expand NBT and other abbreviations in the text

AUTHOR’S RESPONSE: L 254, (page 7 and 8) Full form of NBT has been mentioned as nitroblue tetrazolium and other abbreviations also written first in full form like 1-aminocyclopropane-1-carboxylate deaminase (ACC deaminase) and superoxide dismutase (SOD).

  • line 159 - what are the micro and macro elements?

AUTHOR’S RESPONSE: L 187, (page 5) the content of potassium, nitrogen, iron, and phosphorus was analyzed in the soil and also mentioned in the table 1.

  • line 163 - incorrectly written kg unit; 20 days of incubation ?? or days past inoculation

AUTHOR’S RESPONSE: Corrected of unit has been done in the whole script as ‘kg’. L 202 (page 5), its day's past incubation.

  • lines 377, 239 - all lowercase

AUTHOR’S RESPONSE: L 124 (page 4) written in all lower case ashigh-performance liquid chromatography’.

  • line 196 - the absorbance

AUTHOR’S RESPONSE: L 229 and 213, (page 6) the article has been added ‘the absorbance’.

  • line 212 - no spaces

AUTHOR’S RESPONSE: L 277 (page 8), space has been added between 8.1 and software.

  • line 215 - dot not needed

AUTHOR’S RESPONSE: L 281, (page 8) has been removed.

  • line 221 - no comma, etc.

AUTHOR’S RESPONSE: L 286 (page 8) comma has been added.

  • unclear - from line 223 - description unclear, 253-258 the same

AUTHOR’S RESPONSE:  L 291-292, 324 (page 8 and 9) the lines has been rephrased.

  • Methods- the description of the methods is imprecise, no description of the methods used, e.g. for the determination of PGP activity,

AUTHOR’S RESPONSE: L 100-117, (page 4) Details for analysis of PGP characteristics have been mentioned in the respective section

  • The entire methodology must be corrected.

AUTHOR’S RESPONSE: Details of protocols have been added in the respective sections.

  • -Why in the pot experiment the chromium was used only after 30 days and not immediately when sowing, as in the plates experiment.

AUTHOR’S RESPONSE: The response of inoculation of B. subtilis on germination attributes of B. nigra was measured in germination experiment so to confirm their potential to induce stress tolerance in plants and improve phytoextraction the stress was applied after 30 days of germination to control the impact of other factors and to make sure that these positive changes in plant morphological, physiological, and biochemical attributes were due to bacterial inoculation. It was also reported in literature i.e. induction of stress after germination of seeds.

Waheed, S., Ahmad, R., Irshad, M., Khan, S. A., Mahmood, Q., & Shahzad, M. (2021). Ca2SiO4 chemigation reduces cadmium localization in the subcellular leaf fractions of spinach (Spinacia oleracea L.) under cadmium stress. Ecotoxicology and Environmental Safety207, 111230. https://doi.org/10.1016/j.ecoenv.2020.111230

  • lines 206,222 - what was the plant sample? it is necessary to specify whether they were aerial parts or whole plants.

AUTHOR’S RESPONSE: L 268, (page 7 and 8), in this section (2.13) the metal content in plant roots and shoots was measured separately by following the same protocol so the term ''plant sample'' was used in that section. If we mentioned ''plant sample (root, shoot)'' it’s more confusing. The given formula also clearly indicates that the metal content was measured one by one in roots and shoots of plants.

  • no description of the method for determining the concentrations of metals, are they total or mobile values ?? and what was the pH of the soil. It is known that metal ions become mobile at acidic pH 

AUTHOR’S RESPONSE: L85-88, L 182-195, (page 3 and 5) the details for analysis of soil sample has been mentioned. Before starting the experiment soil pH was measured after inoculation the PGPR modulate the pH of soil when chromium stress was applied. They also increased the mobility of Cr by activating ion channels and different metabolic activities as mentioned in the respective article.

Chen, L.; Luo, S.; Xiao, X.; Guo, H.; Chen, J.; Wan, Y.; Liu, C. Application of plant growth-promoting endophytes (PGPE) isolated from Solanum nigrum L. for phytoextraction of Cd-polluted soils. Appl. Soil Ecol. 2010, 46, 383-389. https://doi.org/10.1016/j.apsoil.2010.10.003

Reviewer 2 Report

The manuscript by Akhtar et al. reports of isolation of a strain of Bacillus subtilis from contaminated soil and the effects of the inoculation on Brassica plants subjected to Cr stress.

The work is extensive, and there are many results shown in the paper and supplementary material. The paper is mostly written in good English but some paragraphs need extensive revision.

I have problems with this paper because:

-the information on methods is never detailed enough and reproducibility would be difficult

-for statistical analysis, the number of replicates are never given

-all results concerning the strain isolation are not shown and the identity of the strain is not demonstrated

-there are no pictures of the plants and no information about their state of health - however there are measurements of many parameters showing stress and improvements

-there is no evidence that the bacteria really survived in the soil and were present during the experiment

For these reasons, even if the presentation and discussion of results are clear, the whole paper is based on data which cannot be considered reliable with the current description.

L20 the correct authority for this species is Brassica nigra (L.) K.Koch after the first citation in the text you should simply use B. nigra  without any other indication

L63-64 recent literature is not so positive about phytoremediation of metals with plants. The authors should check some additional references

L65-66 and others: when taking plant names from the literature, it is important to check the correctness of names in updated databases and report the full name with botanical authority. In this way, the plants can be recognized even if the name is changed. For instance, Thlaspi caerulescens J.Presl & C.Presl can also be known as Noccaea caerulescens

All plant species names should be checked and verified

L67 report the correct name and authority for Brassica nigra, and after this use only B. nigra

L70-72 this is not clear and should be explained better

L95 botanical authority needed

L99 the methods for analysis have not been explained

L104 the concentrations refer to the salt or to the Cr ions, see L114? It would be better to use mol/L

L109-111 These methods are not explained

L113-118 this part is not written in correct English

L115, L144 centrifugation speed must be in xg, as in L123

L129-135 this description is insufficient to describe the method

L141-143 badly expressed

L145-146 what does it mean, primed?

L149 Table 3 cannot be mentioned before Tables 1 and 2

L166-167 this treatment condition is not understandable - how much Cr was given to the pot?

L167 it is not clear how many pots were prepared and how many plants were measured for each treatment

L168 Table 1 has not been cited in the text

***Other methods do not contain sufficient details on solutions, concentrations, reagents, standards and equipment used.

L246: the 20 mg/L Cr stress refers to the concentration of the salt, not of Cr. It is a misleading indication

L267-270 these results are not shown

L279 Table 3: the indication on treatments, also considering the methods' description, is not clear

*** all the results shown give no indication of the number of replicates

L394-399 this part is written in bad English

L456 report botanical authority

L477 botanical authority

Author Response

RESPONSE TO SECOND  REVIEWR’S COMMENTS

  1. Comments and Suggestions for Authors

The manuscript by Akhtar et al. reports of isolation of a strain of Bacillus subtilis from contaminated soil and the effects of the inoculation on Brassica plants subjected to Cr stress.

The work is extensive, and there are many results shown in the paper and supplementary material. The paper is mostly written in good English but some paragraphs need extensive revision.

AUTHOR’S RESPONSE: We are thankful for the valuable suggestions. The necessary changes have been made in the whole manuscript and mentioned accordingly in this draft.

  • the information on methods is never detailed enough and reproducibility would be difficult

AUTHOR’S RESPONSE: Details of different protocols have been added in the sub headings of material and methods.

  • for statistical analysis, the number of replicates are never given

AUTHOR’S RESPONSE: L277-281, (page 8) each treatment has 'three replicates' mentioned in the respective section.

  • all results concerning the strain isolation are not shown and the identity of the strain is not demonstrated

AUTHOR’S RESPONSE: A total of 15 strains have isolated the results regarding characteristics of selected strain (B. subtilis) was given in section 3.2 in L314-315 (page 8), the description of morphology has given in the section 3.3 (page 9).The identity of strain has been established by 16S rRNA sequencing.

  • there are no pictures of the plants and no information about their state of health - however there are measurements of many parameters showing stress and improvements.

AUTHOR’S RESPONSE: (page 9) a picture of seedlings grown in the germination experiment has been added (Figure 1).

  • there is no evidence that the bacteria really survived in the soil and were present during the experiment

AUTHOR’S RESPONSE: The analysis of different parameters have shown that there was a significant difference in morphological, physiological, and biochemical parameters of inoculated and uninoculated plants. The growth of plants in non-stress conditions having bacterial inoculation was also significant as compared to plants grown in control conditions it indicates that PGPR performed their role in stress and un-stress conditions.  After the completion of experiment B. subtilis was isolated from the soil (pots) and identified on the basis of morphological characteristics and CFU was also examined.  

Al-Dhabaan, F. A. (2019). Morphological, biochemical and molecular identification of petroleum hydrocarbons biodegradation bacteria isolated from oil polluted soil in Dhahran, Saud Arabia. Saudi journal of biological sciences26(6), 1247-1252. https://doi.org/10.1016/j.sjbs.2018.05.029

  • For these reasons, even if the presentation and discussion of results are clear, the whole paper is based on data which cannot be considered reliable with the current description.

AUTHOR’S RESPONSE: We have tried our best to answer the queries raised by the reviewer. We look forward to your kind consideration.

  • L20 the correct authority for this species is Brassica nigra (L.) K. Koch after the first citation in the text you should simply use nigra without any other indication

AUTHOR’S RESPONSE: Authority has been corrected as Brassica nigra (L.) K. Koch (L 20, page 1) and after writing full in the whole text ‘B. nigra’ is used.

  • L63-64 recent literature is not so positive about phytoremediation of metals with plants. The authors should check some additional references

AUTHOR’S RESPONSE: L 499-511 (page 17) additional detail has been added about phytoremediation of heavy metals.

  • L65-66 and others: when taking plant names from the literature, it is important to check the correctness of names in updated databases and report the full name with botanical authority. In this way, the plants can be recognized even if the name is changed. For instance, Thlaspi caerulescensPresl & C.Presl can also be known as Noccaea caerulescens

All plant species names should be checked and verified

AUTHOR’S RESPONSE: L 51, (page 2) The botanical name with authority  ‘‘Thlaspi caerulescens (J.Presl & C.Presl) F. K. Mey.’’  has been mentioned. The plants names have been corrected with authority in the whole text.

  • L67 report the correct name and authority for Brassica nigra, and after this use only B. nigra

AUTHOR’S RESPONSE: Correction has been done in the whole text.

  • L70-72 this is not clear and should be explained better

AUTHOR’S RESPONSE: L71-74, (page 3) the line has been rephrased as ‘There is no extensive research work exploring the role of B. nigra on phytoextraction of heavy metals’.

  • L95 botanical authority needed

AUTHOR’S RESPONSE: L 82, (page 3) authority has been given ‘Chenopodium album L.’

  • L99 the methods for analysis have not been explained

AUTHOR’S RESPONSE: L99-117, (page 3 and 4) detail has been given for the analysis of all plant growth-promoting characteristics of the bacterial strain.

  • L104 the concentrations refer to the salt or to the Cr ions, see L114? It would be better to use mol/L

AUTHOR’S RESPONSE: L 154, It was the conc. of chromium salt i.e. K2Cr2O7. That unit was chosen based on the literature. Mentioned on page 20 reference No. 61.

Seleiman, M.F.; Ali, S.; Refay, Y.; Rizwan, M.; Alhammad, B.A.; El-Hendawy, S.E. Chromium resistant microbes and melatonin reduced Cr uptake and toxicity, improved physio-biochemical traits and yield of wheat in contaminated soil. Chemosphere, 2020, 250, 126239. https://doi.org/10.1016/j.chemosphere.2020.126239

  • L109-111, These methods are not explained

AUTHOR’S RESPONSE: L 129-133, section 2.3 (page 4) The method for the analysis of proline produced by bacterial strains has been written in detail.

  • L113-118 this part is not written in correct English

AUTHOR’S RESPONSE: L 150-158, section 2.4 (page 5) the method of DNA sequencing has been rephrased.

  • L115, L144 centrifugation speed must be in xg, as in L123

AUTHOR’S RESPONSE: L 166, section 2.5 (page 5), the centrifugation speed has been written as ‘1000 g’.

  • L129-135 this description is insufficient to describe the method

AUTHOR’S RESPONSE: L138-149, section 2.4 (page 4 and 5), the description of ‘isolation of bacterial DNA’ has been explained.

  • L141-143 badly expressed

AUTHOR’S RESPONSE: L 188-189, section 2.6 (page 6) that line has been rephrased as ‘Pots were filled with 6kg soil and 120 mL of bacterial culture having 108 CFU/mL was inoculated in the soil and 20 days past incubation ten seeds were sown in each pot’.

  • L145-146 what does it mean, primed?

AUTHOR’S RESPONSE: L 168, section 2.5 (page 5), the seeds were dipped in the bacterial culture having 108 CFU/mL the control seeds were also dipped in distilled water. They were dried to their original weight at 28°C after 6 hrs and used in the germination experiment as mentioned in the literature

Abuamsha, R., Salman, M., & Ehlers, R. U. (2011). Effect of seed priming with Serratia plymuthica and Pseudomonas chlororaphis to control Leptosphaeria maculans in different oilseed rape cultivars. European journal of plant pathology130(3), 287-295.

  • L149 Table 3 cannot be mentioned before Tables 1 and 2

AUTHOR’S RESPONSE: L 171, section 2.5 (page 5), that line has been removed.

  • L166-167 this treatment condition is not understandable - how much Cr was given to the pot?

AUTHOR’S RESPONSE: L 204, Three different conc. of Cr i.e. 20, 30, and 40mg/L was used in this experiment and 500mL solution was applied per pot and mentioned in the text.

  • L 167, section 2.6 (page 6), it is not clear how many pots were prepared and how many plants were measured for each treatment

AUTHOR’S RESPONSE: L 202-203, Total 24 pots i.e. 8 treatments × 3 replicates =24 pots. In each pot, 4 plants were maintained after one week of germination.

  • L168 Table 1 has not been cited in the text

AUTHOR’S RESPONSE: L 197, section 2.6 (page 6), table 1 has been mentioned in the text.

  • ***Other methods do not contain sufficient details on solutions, concentrations, reagents, standards, and equipment used.

AUTHOR’S RESPONSE: Details of experiments are added in the section materials and methods.

  • L246: the 20 mg/L Cr stress refers to the concentration of the salt, not of Cr. It is a misleading indication.

AUTHOR’S RESPONSE: In this research, our main focus is on the extraction of Cr from contaminated soil the salt K2Cr2O7 is used as a source of Cr metal but there is a need to talk about Cr, not about the salts so the conc. of salt used as a source of chromium has been mentioned to clear the idea that we are interested in investigating the phytoremediation of Cr. The source of Cr was mentioned in the section material and methods. The same trend used in literature;

Farid, M., Ali, S., Rizwan, M., Yasmeen, T., Arif, M. S., Riaz, M., ... & Ayub, M. A. (2020). Combined effects of citric acid and 5-aminolevulinic acid in mitigating chromium toxicity in sunflower (Helianthus annuus L.) grown in Cr spiked soil. Pakistan Journal of Agricultural Sciences57(2). DOI: 10.21162/PAKJAS/20.9332

Gill, R. A., Zhang, N., Ali, B., Farooq, M. A., Xu, J., Gill, M. B., ... & Zhou, W. (2016). Role of exogenous salicylic acid in regulating physio-morphic and molecular changes under chromium toxicity in black-and yellow-seeded Brassica napus L. Environmental Science and Pollution Research23(20), 20483-20496. https://doi.org/10.1007/s11356-016-7167-2

  • L267-270 these results are not shown

AUTHOR’S RESPONSE: The results of fresh and dry biomass of plant has been given in Table 4 on page 10.

  • L279 Table 3: the indication on treatments, also considering the methods' description, is not clear

AUTHOR’S RESPONSE: We tried to remove the flaws in the description of methods.

  • *** all the results shown give no indication of the number of replicates

AUTHOR’S RESPONSE: The experiment was performed in triplicate i.e. each treatment has three replicates. The data with replicates was used for statistical analysis and it gives mean values of treatments and different letterings indicate significant p<0.05 difference among treatments. That means values were used to describe the results in the text and standard error was also calculated to show the accuracy of the data.

  • L394-399 this part is written in bad English

AUTHOR’S RESPONSE: L 471-479, (page 15), the sentences has been restructured as ‘It results in the production of reactive oxygen species which decrease membrane stability and damage cell organelles. Chromium toxicity also reduced the rate of antioxidant enzyme activities in plants’.

  • L456 report botanical authority

AUTHOR’S RESPONSE: L 547, (page 17), written as Abelmoschus esculentus (L.) Moench.

  • L 477, botanical authority.

AUTHOR’S RESPONSE: L 568, (page 18), written as ‘Brassica napus L’.

Reviewer 3 Report

The manuscript presented by Akthar et al showed the potential of a bacterial strain fo bioremediation of Cr-contaminated soils with Brassica nigra plants.

Although the results are promising, there are some problems that must be solved, especially with the identification of the Bacillus strain. This is not a Bacillus subtilis. Please, check this result and act consequently.

It is not necessary to provide accession number in the abstract. This information should be in the method section. 

There are so many typos and spelling errors. Also, sentences as the ones in lines 234-235 are not necessary. Please, make a thorough revision.

Why was not Chenopodium album plants selected as plant of study? Is B. nigra able to grown in the places where the samples were collected?

Author Response

  1. Comments and Suggestions for Authors

The manuscript presented by Akthar et al showed the potential of a bacterial strain for bioremediation of Cr-contaminated soils with Brassica nigra plants.

Although the results are promising, there are some problems that must be solved, especially with the identification of the Bacillus strain. This is not a Bacillus subtilis. Please, check this result and act consequently.

AUTHOR’S RESPONSE: The study of morphological attributes shows that it represents the characteristics of B. subtilis also reported in the literature (page 22);

Al-Dhabaan, F. A. (2019). Morphological, biochemical and molecular identification of petroleum hydrocarbons biodegradation bacteria isolated from oil polluted soil in Dhahran, Saud Arabia. Saudi journal of biological sciences26(6), 1247-1252. https://doi.org/10.1016/j.sjbs.2018.05.029

Ndeddy Aka, R. J., & Babalola, O. O. (2016). Effect of bacterial inoculation of strains of Pseudomonas aeruginosa, Alcaligenes feacalis and Bacillus subtilis on germination, growth and heavy metal (Cd, Cr, and Ni) uptake of Brassica juncea. International journal of phytoremediation18(2), 200-209. https://doi.org/10.1080/15226514.2015.1073671

It is identified as Bacillus subtilis by 16S rRNA sequencing the details is also mentioned in the link, https://www.ncbi.nlm.nih.gov/nuccore/MW316052 .

  • It is not necessary to provide accession number in the abstract. This information should be in the method section. 

AUTHOR’S RESPONSE: The accession number has been removed from the abstract and mention in the method section L158 (page 5).

  • There are so many types and spelling errors. Also, sentences as the ones in lines 234-235 are not necessary. Please, make a thorough revision.

AUTHOR’S RESPONSE: L 290-292, section 3.2 (page 8), the sentence has been restructured.

  • Why was not Chenopodium album plants selected as plant of study? Is B. nigra able to grow in the places where the samples were collected?

AUTHOR’S RESPONSE: The members of the Brassicaceae family have the potential of phytoextraction so there is a need to explore more potent species for phytoextraction of heavy metals. It can be recommended that the role of Chenopodium album and B. nigra can be studied in comparison in the future to determine which one is more suitable for phytoextraction of heavy metals.

Round 2

Reviewer 1 Report

Thank You for corrections

Author Response

The authors are thankful to the worthy reviewer

Reviewer 2 Report

The authors have made an effort to reply to the comments, but there are still some requests I need to make

The new insertions in methods description contain mistakes in English, in molarity units and should be checked again

All instruments and software mentioned in the methods must be specified with model, brand, city of manufacturer.

I see now that in some tables the number of significant digits for the measure and for the relative error is not the same: e.g. 1.43 +- 0.034 this must be corrected

Updated information about phytoextraction of metals in the introduction has not been included

The legend of new Figure 1 is not sufficiently detailed

Table 3 is very strange: the columns contain the values and the "+-" sign but no error or SD

I appreciate the effort in checking all plant species names, but the new insertions still are missing the botanical authority

The conditions of treatment with Cr as explained at lines 182-184 and at lines 215-218 are still not clear - it is impossible to understand what was done. the authors have tried to explain in their response but the text of the paper is still not clear.

Considering that the plants were treated for a short period and when very small, I think the discussion should explain how relevant this is in view of phytoextraction, where biomass is a key factor.

The authors need to revise carefully all recent additions for the English language.

Author Response

RESPONSE TO SECOND REVIEWER'S COMMENTS

AUTHOR’S RESPONSE: The authors are thankful for the valuable suggestions of reviewers. The manuscript has been improved according to the suggestions. A detailed response to reviewers’ comments is given below.

The authors have made an effort to reply to the comments, but there are still some requests I need to make

The new insertions in methods description contain mistakes in English, in molarity units and should be checked again

All instruments and software mentioned in the methods must be specified with model, brand, city of manufacturer.

AUTHORS RESPONSE: The details of the instruments used in this study have been specified in the section of materials and methods.

I see now that in some tables the number of significant digits for the measure and for the relative error is not the same: e.g. 1.43 +- 0.034 this must be corrected

AUTHORS RESPONSE: The digits in all tables have been corrected and two digits after the decimal are mentioned in all results. (page 6, 8, 9, 10, 12, 13, and 14).

Updated information about phytoextraction of metals in the introduction has not been included

AUTHORS RESPONSE: Information about phytoextraction of metals from recent papers has been added in the introduction, page 3.

The legend of new Figure 1 is not sufficiently detailed

AUTHORS RESPONSE: The detail of treatments has been added in for figure 1 (page 9).

Table 3 is very strange: the columns contain the values and the "+-" sign but no error or SD

AUTHORS RESPONSE: The values of standard deviation have been added in Table 3 (page 9).

I appreciate the effort in checking all plant species names, but the new insertions still are missing the botanical authority

AUTHORS RESPONSE: The botanical authority has been added i.e. Alnus firma Siebold & Zuccarini, Brassica juncea (L.) Czern (page 16 and 22).

The conditions of treatment with Cr as explained at lines 182-184 and at lines 215-218 are still not clear - it is impossible to understand what was done. the authors have tried to explain in their response but the text of the paper is still not clear.

AUTHORS RESPONSE: The sentences have been rewritten about the induction of chromium stress in sections 2.5 and 2.6 (page 5, 6, and 7). The salt used as a source of chromium stress was K2Cr2O7. Chromium stress was given by the application of solution having concentrations of 20 mg/L, 30 mg/L, and 40 mg/L of K2Cr2O7 in different pots.

Considering that the plants were treated for a short period and when very small, I think the discussion should explain how relevant this is in view of phytoextraction, where biomass is a key factor.

AUTHORS RESPONSE: The recent literature has been added in the discussion about the role of PGPR on biomass/phytoextraction potential plants (page 15).

The authors need to revise carefully all recent additions for the English language.

AUTHORS RESPONSE: The whole manuscript has been carefully checked for correct English usage.

Reviewer 3 Report

The authors have improved the manuscript. However, as an expert in bacterial taxonomy and systematics, I cannot agree with the classification of the bacterial strain, nor the reasons given by the authors in their response are appropriate.

The accession number of the strain did not correspond to a Bacillus subtilis strain, and no other information was present to support this assumption. The closest match to this sequence is a Bacillus sp strain and the closest type strain is Bacillus cereus ATCC 14579T. 

The is no need to include in the abstract that the strain was deposited in the NCBI databank. It is ok just to include this information in the methods section.

There are still some typos that the authors must correct before publication. 

Author Response

Response to third Reviewer’s comments

 We highly appreciate the worthy reviewer for his valuable suggestions and critical insight which were helpful in the improvement of the manuscript. The details of changes made in the article according to the valuable suggestions are as follows.

The authors have improved the manuscript. However, as an expert in bacterial taxonomy and systematics, I cannot agree with the classification of the bacterial strain, nor the reasons given by the authors in their response are appropriate.

The accession number of the strain did not correspond to a Bacillus subtilis strain, and no other information was present to support this assumption. The closest match to this sequence is a Bacillus sp. strain and the closest type strain is Bacillus cereus ATCC 14579T. 

AUTHORS RESPONSE:   We are highly thankful to the reviewer for pointing out this basic mistake. We have rechecked the strain and the phylogenetic analysis of strain revealed that it is Bacillus cereus strains (based on homology) and has been mentioned accordingly in the article.

There is no need to include in the abstract that the strain was deposited in the NCBI databank. It is ok just to include this information in the methods section.

AUTHORS RESPONSE: The given statement has been removed from the abstract (page 1).

There are still some typos that the authors must correct before publication.

AUTHORS RESPONSE: The whole manuscript has been carefully checked for typo and grammatical errors.
